# Grape By-Products as Feedstuff for Pig and Poultry Production

**DOI:** 10.3390/ani12172239

**Published:** 2022-08-30

**Authors:** Mónica M. Costa, Cristina M. Alfaia, Paula A. Lopes, José M. Pestana, José A. M. Prates

**Affiliations:** 1CIISA—Centro de Investigação Interdisciplinar em Sanidade Animal, Faculdade de Medicina Veterinária, Universidade de Lisboa, 1300-477 Lisboa, Portugal; 2Laboratório Associado para Ciência Animal e Veterinária (AL4AnimalS), Faculdade de Medicina Veterinária, Universidade de Lisboa, 1300-477 Lisboa, Portugal

**Keywords:** grape by-products, poultry, swine, growth performance

## Abstract

**Simple Summary:**

Grape is one of the most produced fruits worldwide for juice and winery industries. Grape by-products include grape pomace, grape seed, and grape-seed oil, and are valuable, although underexplored, ingredients for pig and poultry feeding. Indeed, they are rich in fiber and bioactive phenolic compounds, which makes them promising sources to partially replace conventional and unsustainable feedstuffs. However, grape by-products are mostly discarded or misused (e.g., landfills) with a negative environmental impact. The present review focuses on the effects of grape by-products on poultry and pig production. Overall, these dietary sources could improve piglet growth when added up to 9% feed, conversely to poultry where this result was only obtained using by-products up to 3%. The beneficial effect on animal growth performance is caused by the presence of nutritional and bioactive compounds with consequent enhancement of intestinal health. The incorporation of high levels of grape by-products in poultry diets can impair growth performance due to the presence of anti-nutritional compounds. Therefore, the use of processes, such as enzymatic supplementation and pre-treatments, to degrade or inhibit these compounds, should be further explored to allow grape by-products to be used as feed ingredients for monogastric animals.

**Abstract:**

Grape by-products are exceptional options for replacement of conventional and unsustainable feed sources, since large amounts are generated every year from the winery industry. However, the majority is wasted with severe environmental and economic consequences. The present review aimed to evaluate the effects of grape by-products on pig and poultry growth performance. The most recent literature was reviewed using ScienceDirect and PubMed databases and the results of a total of 16 and 38 papers for pigs and poultry, respectively, were assessed. Fewer studies are documented for pig, but the incorporation of grape by-products up to 9% feed led to an improvement in growth performance with an increase in average daily gain. Conversely, lower levels (<3% feed) are needed to achieve these results in poultry. The beneficial effects of grape by-products on animal performance are mainly due to their antioxidant, antimicrobial, and gut morphology modulator properties, but their high level of cell wall lignification and content of polyphenolic compounds (e.g., tannin) limits nutrient digestion and absorption by monogastric animals. The use of exogenous enzymes or mechanical/chemical processes can provide additional nutritional value to these products by improving nutrient bioavailability. Overall, the valorization of grape by-products is imperative to use them as feed alternatives and intestinal health promoters, thereby contributing to boost circular agricultural economy.

## 1. Introduction

The steady growth of the human population in the last decades has increased the demand for animal products worldwide, including the most consumed meats (i.e., poultry and pork) and eggs [1]. Feed supply contributes to the highest percentage of total livestock production costs [2]. Therefore, finding sustainable and economically viable alternatives to replace conventional feedstuffs is imperative. Agro-industrial by-products represent an exceptional option of replacement, since they are generated in large amounts every year, mostly from juice and winery industries, the majority being discarded or misused (e.g., landfills) [2]. Indeed, the worldwide wine industry produces thousands of tons of residues per year leading to a big challenge for waste management [3]. Besides their availability and low cost, these by-products are a rich source of nutritional compounds, in which fiber, proteins, minerals, and vitamins, as well other elements, such as antioxidants, stand out. In addition, they can be used as antimicrobial agents, and, thus, reducing the need for antibiotics [2].

Fruit-derived by-products are extensively generated worldwide, and the grape agro-industrial sector is of upmost economic importance [2]. *Vitis vinifera sativa* production is of high relevance because their grape berries, raw and dried, display excellent nutritional components along with pharmaceutical properties, in particular the grape derivatives such as peel and seed extracts [4,5]. In what concerns the wine production, grape pomace accounts for 62%, wine lees account for 14%, and stalk accounts for 12% of organic waste [6]. The main by-product of the grape industry is grape pomace, which contains skins, stems, seeds, and pulp. This product is an intermediate of wine production that is obtained after pressing or crushing whole fruits [2,7]. The global production of wine was 260 million hectoliters in 2021 [8], which contributes to the generation of millions of tons of by-products per year. The Mediterranean countries were among the greatest producers, Italy, France, and Spain standing out with productions of 50.2, 37.6, and 35.3 million hl, respectively, and, with a minor production, Portugal (7.3 million hl) [8]. Altogether, this corresponded to 50.2% of the total amount of wine produced worldwide. However, other countries such as the United States, Australia, Chile, Argentina, and South Africa, were also significant wine producers, with productions of 24.1, 14.2, 13.4, 12.5, and 10.6 million hl of wine in 2021 (28.9% of total) [8]. Since the beginning of the 21st century, an increased awareness for the importance of adopting a sustainable agriculture and livestock practice has occurred [9]. The need to reduce the environmental impact of waste disposal gives additional value to grape by-products for several industrial applications, such as soil fertilization, feed, food, bioenergy (biogas), biofuel (bioethanol), pharmaceutical, nutraceutical, and cosmetic compounds [10,11]. The potential of by-products for composting and fertilization is remarkable since they might present suitable organic loads and mineral elements for plant growth, although it is necessary to control the levels of minerals that can act as contaminants [12]. Moreover, the incorporation of these sources into the animal diet requires an evaluation of their effect on animal production, by testing their outcomes, positive or negative, on growth performance. Climate changes are one of the most important problems that agriculture faces nowadays. The future relies on how to continue livestock production, mitigating the negative effects for the farmer. In general, climate change has high economic, environmental, and social costs, but the sectors most influenced and affected are agriculture and forests, as they develop in the open air, and depend almost solely on weather conditions. The environmental impact of humankind results in modifications of water resources and irrigation requirements, soil fertility, salinity, and erosion, crop growth conditions, productivity and distribution, land use, optimal conditions for livestock production, agricultural pests and diseases, and increased expenditure on emergency and remediation actions [3].

The present review updates the results of using grape by-products from the agro-industrial sector, as one of the main crops with largest expression worldwide, on the production performance of pigs and poultry.

## 2. Materials and Methods

The methodology used was based on the search literature from the last decades, covering ScienceDirect (Elsevier, Amsterdam, The Netherlands), Web of Science (Clarivate Analytics, Philadelphia PA, USA), and PubMed (NCBI, Bethesda MD, USA) databases, and using “grape”, “by-products”, “grape by-products”, “growth performance”, hens”, “laying hens”, “laying hen pullets”, “turkeys”, “broiler”, “broiler chicken”, “quails”, “weaned pigs”, “weaning”, “grape pomace”, “seedless pomace”, “grape seed”, “poultry”, “weaned piglets”, and “pigs” as keywords. No papers were found for “laying hen pullets”, “turkeys”, and “seedless pomace”. The literature search was performed in June of 2022 and a total of 16 and 38 papers were found for pigs and poultry, respectively. 

## 3. Nutritional Properties of Grape By-Products and Strategies to Overcome Their Negative Effects

The processing of grapes (*V. vinifera*) into wine or juice generates a large number of solid residues that correspond to several by-products, including vine shoots, stalks, pomace, grape-seed extract, lees, and spent filter cakes. These are good sources of bioactive compounds [13], which makes them suitable to be considered as functional ingredients [14]. The chemical composition of grape by-products depends on the maturity level, environmental factors, grape variety [15], and the technology employed in the wine-making process [2]. Nevertheless, they are generally rich in phenolic compounds and fiber (Table 1 and Table 2). 

Grape pomace accounts for about 20–25% of the grape weight used for wine production [16], and, thus, is predominantly rich in a wide range of polyphenols, particularly located in the skin [13], but also contains fiber, proteins, lipids, and minerals. The protein (12% dry weight) contains the essential amino acids for monogastric animals, although tryptophan was not measured in the reported studies. There is a predominance of arginine (7.2%), aspartic acid (7.1%), glutamic acid (11.5%), glycine (6.2%), and threonine (21.7%). Lysine, which is the first limiting amino acid in pig and the second in poultry diets, is present in considerable amounts (average of 4.5% of total amino acids) conversely to methionine (up to 1.4%), the first limiting amino acid for poultry (Table 1). The grape pomace can be divided into two fractions: seedless pomace (residual pulp, stem, and skin) (48–62%) and seeds (38–52%). The first is rich in dietary fiber, whereas seeds are mainly valued for their oil containing unsaturated fatty acids (oleic and linoleic acids). Although the phenolic compound profile is variable, both fractions are rich in flavonoids, such as anthocyanins and proanthocyanidins in seedless pomace and flavonols in seeds [5]. Overall, phenolic compounds present in grape pomace include phenolic acids and alcohols, flavonoids, such as proanthocyanidins and catechin, and stilbenes (resveratrol) [2,13]. Their antioxidant activity has been reported [2,13,17], together with the ability of grape pomace to increase vitamin E synthesis in the liver [18] and glutathione peroxidase and superoxide dismutase activities in the digestive tract of monogastric animals [19]. Although different bioactivities have been attributed to grape pomace fractions, as, for example, a stronger bactericidal effect for seedless pomace, the whole grape pomace is normally used in animal studies [5].

Grape seed is composed of fibers (47%, 60–70% non-digestible), complex carbohydrates (29%), fat (13%, rich in essential fatty acids), proteins (11%), minerals, and extractable phenolic compounds (Table 1 and Table 2). The latter include mostly phenolic acids (e.g., gallic acid) and flavonoids (e.g., protocatechuic acid and epicatechin) [16,20]. The protein is composed of all the essential amino acids, with an average of 4.5% for lysine and 3.5% for methionine. The most predominant amino acids are the same as for grape pomace, except for threonine (4.0%). In fact, glutamic acid and glycine can reach values up to 30.3 and 16.0%, respectively (Table 1). Grape-seed extract and grape-seed oil are two by-products derived from grape seeds. The grape-seed extract is obtained when grape seeds from grape juice or wine processing are extracted, dried, and purified in order to produce a residue enriched in polyphenols [21]. These compounds have strong antioxidant (efficient removal of free radicals with reduction of deleterious oxidative reactions) [22,23] and antimicrobial [20] properties.

The incorporation of grape by-products in animal feeds presents concerns related to the uptake of toxic elements by the grape vineyard from residual biosolids, environmental pollution, and irrigation water [24,25]. These elements include heavy metals (e.g., Al, As, Pb, Cd, and Ni) [26] and toxins, such as ochratoxin A and biogenic amines [11], which are mostly released from chemical pesticides and fertilizers to the cultivation soil. In addition, they can be a result of industrial activities and human traffic [24,25]. Few studies reported the presence of heavy metals in the whole grape pomace [27,28], with values (mg/kg DM) averaging 111 for Al [28], 16.3 for Pb, 3.5 for Ni, and 1.1 for Cd [27]. Indeed, most reports analyzed the accumulation of trace elements in grape stem [27,29], skin [30,31,32], or seed [26,32]. There is a great variability in heavy metal contents in grape by-products, since they depend on soil composition and contamination and on the grape genetic variety. For instance, 8.1 mg/kg DM of Al and 0.002 mg/kg DM of Pb was found in grape skin in the study from De Nisco et al. [30], but other grape varieties could accumulate 158–575 mg/kg DM of Pb in their skins [32]. However, in grape stem and seed, the predominant trace element was Al (average of 201 and 0.78 mg/kg DM, respectively) (Table 2). Although high levels of these trace elements can be harmful to animals with potential carcinogenic effects [33] and kidney, nervous, and immune system toxicity [34], there is a lack of reference values for the evaluation of toxic effects of grape by-products used as feedstuffs [11]. Nevertheless, As, Cd, and Pb should not be higher than 0.5, 0.5, and 0.2 mg/kg in food ingredients [35], and, thus, Pb values, which can reach up to 26.2 mg/kg in grape stem, should be monitored. The minimization of anthropogenic sources of trace elements [25], selection of grape varieties [29], and application of bioremediation techniques to remove heavy metals from the soil [36], are possible solutions to face the concerning issues.

The potential value of by-products in animal feeding depends essentially on their nutritive properties, such as fibrous, protein, and organic-matter digestibility and energy value. Although grape pomace, grape seeds, and grape-seed oil contain beneficial compounds for monogastric animals’ metabolism and growth, such as essential fatty acids and antioxidant phytosterols and tocopherols (vitamin E) (Table 2), these by-products are also composed of anti-nutritional compounds. Indeed, they contain a high amount of fiber and procyanidins (i.e., condensed tannins) and, in a lower quantity, phytic acid [2]. The highest proportions of dietary fiber (74% wt.), mostly composed of hemicelluloses and covered in a whitish bloom (a dusting of wild yeasts and bacteria) [37], and polyphenols (i.e., tannins) [38] are located in the grape skin. However, grape stem is woody and fully composed of tannins, containing more than 50% of total polysaccharides and, thus, representing an economically attractive source of fiber material [39]. However, depending on the dose and treatment of grape by-products included in animal feed, dietary fiber and polyphenolic compounds can maintain or improve pig [40,41,42] or poultry [2] growth performance and health. In fact, fiber increases intestinal peristalsis and acts as a buffer and a prebiotic that stimulates the development of beneficial bacteria in the digestive tract [2], whereas polyphenols can act as antioxidants, antimicrobials, and immunomodulators [20,40,41,42,43,44,45].

Enzyme supplementation or pre-treatment methods, such as fermentation [2], polyethylene glycol treatment [46,47], steam explosion, and amination are possible solutions to release non-starch polysaccharides or linked tannins from grape by-product cell walls and biomass, thereby increasing their digestibility and bioactive properties [2]. Moreover, phytases might be used to hydrolyze phytate and, thus, release organically bound phosphorus that is unavailable for digestion and absorption by monogastric animals [48]. These treatments could increase the use of grape by-products in animal diets with concomitant benefits on animal production and health, reduction in the overexploitation of conventional feedstuffs, feed costs, and environmental impact derived from the discard or incineration of by-products. 

The enzyme supplementation of grape by-products is scarcely studied and has controversial results, since the outcome depends on the by-product dose [2]. For instance, the ability of a carbohydrase complex, and especially tannase, to degrade polymeric procyanidin structures into monomeric and dimeric residues of catechin when acting on 5% and 10% of grape pomace was demonstrated, but the antimicrobial effect of these phenolic compounds against *Clostridium perfringens* in the intestine of broiler chickens was only present with 5% of substrate [49]. However, the activity of exogenous enzymes in the context of pre-treatments, such as fermentation, or in combination with polyethylene glycol, has been applied more and was shown to increase the antioxidant and antimicrobial bioactivities of grape compounds and hinder anti-nutritional effects [2]. Particularly, the fermentation process can increase the amount or effectiveness of bioactive polysaccharides, polyphenols, or mannoproteins [2]. Grape seed fermented with *Aspergillus niger*, which produces several enzymes such as amylase, protease, xylanase, cellulase, and lipase, was shown to modulate intestinal microbiota of broiler chickens causing an increase in beneficial *Lactobacillus* and a decrease in *Staphylococcus aureus* [20]. This process was also efficient in the bioconversion of grape pomace, stimulating both antioxidant (increase in serum catalase level) and microbial modulation (decrease in *C. perfringens*) effects of grape-derived compounds [44]. Moreover, the pre-treatment of 10% red grape pomace with polyethylene glycol, which is a strong tannin-binding agent [50], and a cellulolytic enzyme mixture was shown to ameliorate the anti-nutritional effects of condensed tannins [46]. In addition, Van Niekerk and Mnisi [47] reported a partial inactivation of grape pomace condensed tannins with polyethylene glycol, without affecting the health status of broiler chickens. Concerning the hydrolysis of phytate phosphorus, few studies evaluated the activity of phytase in grape by-products, whether using enzymes naturally present in the substrate [48] or biosynthesized by *A. niger* during a fermentation process [51]. This might be due to the residual amounts of phytic acid usually present in the by-products [48,52] and an inhibition of fermentation by phenolic compounds, such as resveratrol and anthocyanins, with anti-fungal activities [52].

The amination nd steam explosion treatments have not been exploited for grape by-products, although they are known to increase the digestibility of fibrous cell walls in high-fiber feedstuffs [2]. Amination breaks down hemicellulose and lignocellulose and enhances available nitrogen and soluble sugar contents by using chemical compounds (e.g., ammonia), whereas steam explosion is also effective as a lignocellulose material treatment through the application of high temperature and pressure, but it does not increase nitrogen content [53,54]. Even though steam explosion has more advantages than amination, since it is cost-effective, does not use chemicals, and has low energy expenditure [2], both methods could be used for delignification of cell walls in grape pomace, solving the problem of the high level of lignification of this by-product that compromises monogastric animals’ digestibility.

**Table 1 animals-12-02239-t001:** Metabolizable energy, protein content, amino acid profile, and carbohydrate content of grape pomace and grape seed (values are expressed on a dry weight basis, w/dw, hyphenated values are ranges).

Item	Grape Pomace ^1^	Grape Seed ^2^
Moisture (%)	3.39–10.2 (7.2)	4.95–7.60 (5.71)
Metabolizable energy (MJ/kg)	5.1–8.7 (6.7)	4.7–6.9 (5.9)
Crude protein (%)	8.9–13.9 (12.1)	6.0–16.7 (11.2)
**Amino Acid Profile (% Total Amino Acids)**		
Alanine	3.3–5.4 (4.2)	2.3–7.4 (5.6)
Arginine	6.2–8.0 (7.2)	6.1–8.2 (7.3)
Aspartic acid	5.6–8.4 (7.1) ^3^	5.0–8.4 (7.1)
Cystine/Cysteine	0.06–0.39 (0.18)	2.4–2.5 (2.5)
Glutamic acid	9.0–13.6 (11.5) ^4^	3.6–30.3 (18.9)
Glycine	6.1–6.3 (6.2)	3.5–16.0 (10.2)
Histidine	2.8–4.0 (3.2)	2.3–3.4 (2.9)
Isoleucine	2.9–4.8 (3.5)	2.5–3.5 (3.1)
Leucine	5.2–7.7 (5.9)	2.7–5.8 (4.8)
Lysine	2.3–8.3 (4.5)	3.9–5.3 (4.5)
Methionine	0.51–1.4 (0.76)	3.5–3.6 (3.5)
Phenylalanine	4.5–5.1 (4.7)	2.6–12.2 (5.4)
Proline	4.8–8.8 (5.9)	2.7–6.8 (4.8)
Serine	3.6–5.4 (4.3)	4.9–6.9 (5.9)
Threonine	5.3–33.1 (21.7)	3.2–5.0 (4.0)
Tryptophan	n.a. ^5^	4.7–5.4 (5.0)
Tyrosine	3.0–4.1 (3.6)	1.7–6.1 (3.4)
Valine	3.5–6.0 (4.3)	3.2–5.0 (4.1)
**Crude Carbohydrates (%)**		
Crude fiber	14.3–74.5 (38.9)	45.8–47.4 (46.6)
ADF	32.3–48.4 (40.4)	ND
TDF/NDF	40.9–59.1 (48.8)	40.8–58.2 (47.7)
SDF	2.4–9.8 (6.1)	ND-79.9
ADL/Lignin	18.2–42.5 (29.8)	ND
Sugar (%)	2.1–14.2 (6.4)	ND

ADF, acid detergent fiber; TDF, total dietary fiber; NDF, neutral detergent fiber; SDF, soluble dietary fibe; ADL, acid detergent lignin; ND, not detected. Supporting literature: ^1^ Alameldin [55]; Atalay [56]; Beres et al. [57]; Chikwanha et al. [28]; Ebrahimzadeh et al. [58]; Ebrahimzadeh et al. [59]; Erinle et al. [45]; Goñi et al. [18]; Gülcü et al. [60]; Gungor et al. [44]; Hosseini-Vashan et al. [61]; Jonathan and Mnisi [62]; Leal et al. [29]; Llobera and Cañellas [63]; Mirzaei-Aghsaghali et al. [64]; Pérez Cid et al. [32]; Valiente et al. [65]; Vlaicu et al. [66]; Winkler et al. [67]; Yi et al. [68]. ^2^ García-Rodríguez et al. [69]; Goñ et al. [70]; Karaman et al. [71]; Milićević et al. [26]; Pérez Cid et al. [32]; Spanghero et al. [72]; Tangolar et al. [73]; Tangolar et al. [74]; Yokotsuka and Singleton [75]. ^3^ Includes minor amounts of asparagine. ^4^ Includes minor amounts of glutamine. ^5^ n.a., not available.

**Table 2 animals-12-02239-t002:** Lipid content and fatty acid profile, and ash, mineral, vitamin E, and phenolic compound content of the main grape by-products (values are expressed on a dry weight basis, w/dw, hyphenated values are ranges).

Item	Grape Pomace ^1^	Grape Seed ^2^	Grape-Seed Oil ^3^
Crude fat (%)	2.12–13.5 (7.9)	4.82–20.7 (12.9)	-
**Fatty Acid Profile (% Total Fatty Acids)**			
16:0	12.0–18.0 (15.5)	7.61–10.0 (8.72)	6.50–9.70 (8.26)
18:0	4.31–7.95 (6.18)	3.14–4.96 (3.87)	2.84–7.30 (4.49)
20:0	0.57–0.84 (0.71)	0.04–0.10 (0.07)	0.14–0.16 (0.15)
16:1*n*-7	0.05–0.10 (0.08)	0.07–0.32 (0.17)	0.08
18:1*n*-9	12.2–28.1 (20.7)	13.6–22.9 (18.3)	14.3–26.5 (20.4)
20:1	0.03–0.04 (0.04)	0.03–0.17 (0.09)	0.00–0.97 (0.39)
18:2*n*-6	43.2–62.7 (52.1)	62.5–73.8 (67.9)	60.1–74.7 (66.0)
18:3*n*-3	0.12–2.80 (1.00)	0.21–0.35 (0.29)	0.00–0.87 (0.42)
SFA	20.6–21.8 (21.2)	12.0–15.1 (13.3)	10.4–11.7 (13.1)
*cis*-MUFA	14.3–15.4 (16.5)	18.2–23.3 (20.0)	14.8–18.7 (16.7)
PUFA	60.9–64.4 (62.7)	62.9–69.5 (66.6)	68.3–74.9 (71.6)
*n*-3 PUFA	1.70–2.80 (2.25)	0.16–0.35 (0.27)	0.20
*n*-6 PUFA	58.1–62.7 (60.4)	62.5–69.2 (66.3)	74.7
Ash (%)	2.4–23.7 (6.9)	2.60–20.1 (13.0)	
Mineral composition			
Macrominerals (g/kg)			
Ca	3.20–4.70 (4.00)	4.80–7.90 (6.95)	-
K	8.99–33.1 (20.3)	3.30–8.91 (5.05)	-
Mg	0.80–1.32 (1.08)	1.30–1.87 (1.61)	-
P	2.4–23.8 (13.7)	0.83–29.6 (9.12)	-
Microminerals (mg/kg)			
Al	46.8–496 (201) ^4^	0.78	-
As	0.11–0.79 (0.33) ^4^	0.0019	-
Cd	0.004–0.8 (0.12) ^4^	0.0009	-
Cu	12.4–387 (115)	7.27–28.0 (13.5)	-
Fe	94.3–109 (103)	17.3–54.0 (28.5)	-
Hg	0.012–0.022 (0.017) ^5^	0.006–0.016 (0.012)	-
Mn	16.2–60.0 (27.6)	11.1–27.5 (18.3)	-
Ni	8.7 ^4^	0.076	
Pb	0.02–26.2 (3.79) ^4^	0.001–0.16 (0.068)	-
Zn	11.7–18.8 (14.6)	12.3–26.9 (16.1)	-
**Vitamin E Homologues (mg/kg)**			
α-Tocopherol	3.3–6.4 (4.9)	2.99–3.35 (3.18)	162–578 (430)
β-Tocopherol	2.3–6.7 (3.8)	2.55–2.73 (2.66)	-
γ-Tocopherol	3.8–9.8 (6.9)	8.48–12.8 (10.9)	20.0
δ-Tocopherol	1.9–2.2 (2.0)	ND-3.04	1.00
Phenolic compounds			
Total anthocyanin (mg/g)	0.53–3.4 (1.6)	ND	-
Total flavonoids (mg CE/g)	15.4–26.9 (21.0)	34.6–36.7 (35.8)	-
Total phenols	12.3–58.9 (27.9) ^7^	48.0–120 (93.7) ^6,7^ 261–363 (312) ^8^	-
Total tannins	96.9–139 (114) ^9^	125–127 (126) ^9^ 33.9–56.3 (45.1) ^8^	-

SFA, saturated fatty acids; MUFA, monounsaturated fatty acids; PUFA, polyunsaturated fatty acids; GAE, gallic acid equivalents; TAE, tannic acid equivalents; CE, catechin equivalents; ND, not detected. Supporting literature: ^1^ Alameldin [55]; Beres et al. [57]; Bustamante et al. [27]; Ebrahimzadeh et al. [59]; Erinle et al. [45]; Goñi et al. [18]; Gülcü et al. [60]; Gungor et al. [44]; Hosseini-Vashan et al. [61]; Jonathan and Mnisi [62]; Kasapidou et al. [76]; Leal et al. [29]; Llobera and Cañellas [63]; Pérez Cid et al. [32]; Vlaicu et al. [66]; Yi et al. [68]. ^2^ Goñi et al. [70]; Gülcü et al. [60]; Gungor et al. [20]; Iuga and Mironeasa [15]; Karaman et al. [71]; Milićević et al. [26]; Pérez Cid et al. [32]; Silva et al. [77]; Tangolar et al. [73]; Yokotsuka and Singleton [75]. ^3^ Baydar and Akkurt [78]; Bravi et al. [79]; Duba and Fiori [80]; Fernandes et al. [81]; Karaman et al. [71]; Orsavova et al. [82]. ^4^ Values were detected in grape stem. ^5^ Values were detected in grape skin. ^6^ More than 99.5% of the total phenols are flavonoids. ^7^ Values are expressed as mg GAE/g. ^8^ Values are expressed as mg epicatechin equivalents/g. ^9^ Values are expressed as mg TAE/g.

## 4. Effect of Dietary Grape By-Products on Production Performance of Monogastric Animals

The literature review on the influence of dietary grape by-products on growth performance parameters of monogastric species is presented in Table 3 and Table 4. The studies herein presented used grape by-products at up to 10% feed with variable effects for pigs and poultry. In general, although different concentrations and experimental periods were reported, dietary grape by-products did not impair or even improved growth performance with an increase in average daily gain (ADG) in pigs, with positive results for poultry when added in low amounts (<3% feed) in the diet (Figure 1). The effects on animal growth performance were mostly attributed to the bioactive properties of grape by-products compounds (e.g., polyphenols), which include prevention of oxidative stress, as well as immune, microbiota, and gut morphology modulation, in pigs [40,41,42,83] and poultry [20,43,44,45]. For instance, Hao et al. [40] and Fang et al. [41] showed that procyanidins added at up to 1.5 and 1% in piglet diets increased glutathione peroxidase and superoxide dismutase activities in the serum and liver, respectively. Fang et al. [41] reported an increase in serum immunoglobulins, interleukins, and complements caused by these compounds. In addition, polyphenols could improve the disease resistance in piglets by enhancing the proportion of beneficial intestinal bacteria (e.g., *Lactobacillus* sp., *Olsenella* sp., *Selenomonas* sp.) [42] and decreasing the incidence of diarrhea [40,41]. The bioactive compounds were also responsible for an increase in intestinal villus height/crypt depth ratio when grape pomace was fed at 5% to piglets [42]. This bioactivity might explain the increase in average daily gain (ADG) and decrease in feed conversion ratio (FCR) [41] or the ameliorated effects [40,42] on growth performance in piglets fed grape by-products.

Similar bioactivities of grape compounds were detected in poultry, but the outcomes were shown to be dependent on by-product dose, treatment, and type. Indeed, 1.5% of grape pomace fed to broiler chickens increased serum glutathione peroxidase and superoxide dismutase levels, had no effect on intestinal bacteria count, and enhanced ileum lamina thickness. On the other hand, 1.5% of fermented grape pomace increased serum catalase, decreased cecal *C. perfringens*, and did not change ileal morphology [44]. Furthermore, Viveros et al. [43] reported an increase in ileal *Lactobacillus* with 0.72% of grape-seed extract, and an increment in *Enterococcus* and decrease in *Clostridium* with grape-seed extract and 6% of grape pomace fed to broilers. Grape pomace increased intestinal villus height/crypt depth ratio, but an opposite effect occurred when feeding grape-seed extract. Additionally, Erinle et al. [45] found an increase in beneficial *Lactobacillus* and improvement in gut morphology in broilers fed 2.5% of grape pomace. Overall, the bioactivity of grape compounds was variably associated with an increase in final body weight [44] and ADG [20] and decrease in FCR [43].

### 4.1. Pigs

The effect of dietary incorporation of grape by-products on pig growth performance is mostly dependent on animal growth stage and by-product dose on a dry basis. In general, lower concentrations of grape by-products had fewer effects on growth performance, although depending on the concentrations applied and on the age of the animal. For instance, considering animals with similar weights (4.8–19.3 kg), pigs fed 9% of grape pomace had an increase in ADG during all experiments, while those fed 5% suffered no change in ADG (Table 3). The piglets fed 3% of the by-product improved their growth performance only during the growing stage (36–70 days) possibly due to increased digestibility [84]. Beside the differences in trial conditions, the pre-treatment of grape pomace should be considered, since the by-product used in the study by Yan and Kim [84] was submitted to a fermentation process, and, thus, contained a high phenolic content (62.1 g/mg). Moreover, Kafantaris et al. [85] observed that 9% of grape pomace fed to piglets for 30 days increased ADG, without affecting average daily feed intake (ADFI) and FCR. In growing-finishing pigs, Trombetta et al. [86] reported no effects on ADG and carcass traits with 3.5 and 7% of grape pomace. Additionally, the supplementation of 5% of flax meal and 1% of grape seeds did not affect ADG and FCR, although increased ADFI [66]. A lack of a significant effect of grape-seed extract on these parameters was also observed when piglets were fed 0.015% of the by-product [83]. However, a high dose of grape by-product might not always influence growth performance in pigs, since piglets born from sows fed with grape pomace (25% feed) and fed with the same ration before weaning, showed no difference in ADG and FCR [87]. Even though the results obtained across studies are variable, the increase in ADG demonstrated in some reports should be underlined, since it is a promising effect for animal production.

**Table 3 animals-12-02239-t003:** Literature review on the effects of dietary inclusion of grape by-products on production performance of pigs.

Grape By-Product	Level in the Diet (% Dry Weight) and Experiment Duration	Animal and Initial Weight/Age	Main Findings	References
Grape pomace (fermented)	3% for 105 days	Pigs with 19.3 kg	No effect on ADFI, final body weight, ADG, FCR, and *longissimus* muscle area Increase in ADG (36–70 days)	Yan and Kim [84]
Grape pomace	5% for 36 days	Piglets with 10.7 kg	Increase in ADFI (d16–36 and overall period) No effect on ADG and FCR	Chedea et al. [88]
	9% for 30 days	20-day-old piglets with 4.8 kg	No effect on ADFI and FCR Increase in final body weight and ADG	Kafantaris et al. [85]
	5% for 24 days	Fattening-finishing pigs	No effect on ADG and ADFI	Taranu et al. [89]
	3.5 and 7% for 86 days	180-day-old castrated males and female pigs with 48.6 kg	No effect on ADG, hot carcass yield, loin area, and backfat thickness	Trombetta et al. [86]
	5% for 28 days	28-day-old piglets	No effect on ADG, ADFI, and FCR	Wang et al. [42]
Grape pomace (spent grapes)	Replacement of 25% maize for 63 days	Pregnant sows (3 weeks up to weaning) with 180.5 kg and their piglets	No effect on final body weight, ADFI (sows and piglets), ADG, and FCR (piglets)	Tripura et al. [87]
Grape-seed extract (procyanidins)	0.04, 0.07, and 1% for 28 days	Piglets (mixed sex) with 8.4 kg	Increase in ADG and decrease in FCR (0.04% dosage)	Fang et al. [41]
	0.5, 1, and 1.5% for 28 days	21-day-old piglets with 6.99 kg	No effect on ADG, ADFI and FCR	Hao et al. [40]
Grape-seed extract	0.015% for 56 days	Piglets (mixed sex) with 6.9 kg	Decrease in final body weight (d13) in relation to in-feed antibiotic treatment No effect on ADFI, ADG, and FCR	Rajković et al. [83]
Grape-seed and grape-marc extracts	1% replacing wheat for 28 days	5-week-old pigs with 10 kg	No effect on ADFI, final body weight, and ADG Decrease in FCR (tendency)	Fiesel et al. [90]
	1% for 28 days	6-week-old pigs with 12 kg	No effect on ADFI, final body weight, and ADG Decrease in FCR	Gessner et al. [91]
Grape-seed cake	5% for 24 days	Pigs with 75.5 kg	No effect on ADG and ADFI	Taranu et al. [92]
Grape seeds	8% for 30 days	Piglets with 9.13 kg	No effect on ADFI, final body weight, ADG, and FCR	Grosu et al. [93]
	1% of grape seeds and 5% of flax meal from 65.3 to 105 kg	Pigs with 60.3 kg	No effect on final body weight, ADG, and FCR Increase in ADFI	Vlaicu et al. [66]
	8% for 30 days	Piglets with 9.13 kg	No effect on final body weight and ADG	Taranu et al. [94]; Taranu et al. [95]

ADG, average daily gain; ADFI, average daily feed intake; FCR, feed conversion ratio.

### 4.2. Poultry

Poultry feeding with grape by-products showed both dosage- and form-dependent effects, and, thus, they are normally incorporated at up to 6–10% feed (dry basis) in the diet [2]. Indeed, the inclusion of high levels of grape by-products is normally associated with a considerable amount of anti-nutritional compounds, such as fiber and polymeric polyphenols (e.g., proanthocyanidins), which can reduce the digestion and absorption of nutrients, and ultimately impact negatively on body weight gain. On the other hand, low levels of by-products (<6% feed) have been related to beneficial bioactive effects, such as modulation of gut morphology and microbiota and antioxidant activity, mostly due to the presence of polyphenols [2]. For instance, feeding broilers with 2.5% of grape pomace increased the relative abundance of beneficial *Bacteroides* and *Lactobacillus* genera and reduced the *Firmicutes* to *Bacteroidetes* ratio in the cecum [45]. Erinle et al. [45] also demonstrated an increase in intestinal villus height: crypt depth ratio caused by the grape by-product. In addition, Abu Hafsa and Ibrahim [96] reported that 1–4% of grape seeds increased *Lactobacillus* and decreased detrimental *Streptococcus* spp. and *Escherichia coli* in the ileum of broilers. Similarly, Viveros et al. [43] found an increase in *Lactobacillus* sp. in the ileal contents of broiler chicks fed 0.72% of grape-seed extract. Dietary grape seeds at up to 4% can help prevent oxidative stress by increasing the activity of several enzymes with antioxidant activity in the plasma, such as superoxide dismutase and glutathione peroxidase, and, at the same time, reduce thiobarbituric-acid-reactive substances [96]. Overall, the bioactivity of grape by-products contributes to the integrity of intestinal barrier function and prevention of diseases in the gut, thereby enhancing poultry growth [2]. 

The inclusion of grape pomace at 6% in broiler chicken diets showed no effects [97] or a slight improvement [43] on growth performance. However, a negative impact on performance was observed when the same dose of grape-seed extract [43] or unfermented grape skin [38] was used, which was due to the presence of polyphenols, mostly purified in the case of grape seed. Feeding broilers with a lower dose grape seed (4%), which corresponds to 11.14 g polyphenols/kg feed [96], could still impair growth performance by decreasing final body weight and ADG and increasing FCR. Similarly, a linear increase in FCR over the growing and finishing periods of ducklings was observed with 0.01 and 0.02% of grape-seed extract providing 0.005–0.015 g polyphenols/kg feed, although an increase in final body weight and ADG was reported [98].

For a level of by-products higher than 6%, only Kumanda et al. [99] reported beneficial effects on broiler growth performance with a decrease in FCR when incorporating 7.5% of grape pomace in the diet. Other studies showed either no effect [59,100] or an impairment (decrease in final body weight, ADG, and hot carcass weight) [46] of animal performance with 10% grape pomace. 

On the other hand, studies showed that the inclusion of grape seed or grape pomace at up to 3% feed (dry basis) had a positive impact on animal growth performance. Indeed, 0.5% of fermented or unfermented grape seed fed to broilers increased their final body weight and ADG [20] and even grape-seed extract at 2% feed led to similar results with also a reduction in FCR [96]. In addition, 1.5% of fermented grape pomace could increase the final body weight of broilers [44], whereas 2.5% of grape pomace raised ADG in the first two weeks of the trial, even though no effect was observed in the overall period [45].

The fermentation of grape by-products was reported to hinder the negative effects on growth performance caused by the presence of anti-nutritional compounds [2]. However, this occurrence was mostly shown for low levels of by-products in broiler diets. For instance, Nardoia et al. [38] observed that the pre-treatment reverted the increase in FCR found with unfermented grape skin at 3%. Moreover, Gungor et al. [44] reported a beneficial effect on animal growth with the fermentation of 1.5% grape pomace, while the same was not found with the unfermented source.

**Table 4 animals-12-02239-t004:** Literature review on the effects of dietary inclusion of grape by-products on production performance of poultry.

Grape By-Product	Level in the Diet (% Dry Weight) and Experiment Duration	Animal and Initial Weight/Age	Main Findings	References
Grape pomace	0.5, 0.75, and 1% for 28 days	3-day-old broiler chicks	No effect on ADG, ADFI, FCR, and carcass weight	Aditya et al. [101]
	1.5, 3, and 6% from 21 to 42 days	21-day-old male broiler chicks	No effect on growth performance	Brenes et al. [97]
	5 and 10% for 21 days	1-day-old male broiler chicks	No effect on final body weight, ADFI, and FCR	Chamorro et al. [100]
	0.045, 0.035, and 0.025% body weight for 40 days	1-day-old mixed-sex broiler chicks	No effect on final body weight	Dupak et al. [102]
	10% (combined or not with 0.1 or 0.05% tannase) for 42 days	1-day-old male broiler chicks	Decrease in body weight and ADG on d10	Ebrahimzadeh et al. [103]
	5, 7.5, and 10% for 42 days	1-day-old male broiler chicks	No effect on growth performance	Ebrahimzadeh et al. [59]
	2.5% for 42 days	1-day-old mixed-sex broiler chicks	No effect on ADG and FCR but increase in ADFI (overall) Increase in ADG and ADFI (d1–14)	Erinle et al. [45]
	2, 4, and 6% for 84 days	42-day-old quails	No effect on ADFI and FCR and egg production Egg weight linearly increases with dosage	Fróes et al. [104]
	0.5, 1.5, and 3% for 3 weeks	1-day-old male broiler chicks	No effect on ADG, ADFI, and FCR	Goñi et al. [18]
	2, 4, and 6% for 42 days	1-day-old male broiler chicks	No effect on ADG and FCR Increase in ADFI (2, 4 and 6% dosages)	Hosseini-Vashan et al. [61]
	1.5, 3.0, 4.5, and 6.0% for 77 days	5-week-old male cockerels	No effect on ADFI, final body weight, ADG, and FCR	Jonathan and Mnisi [62]
	4 and 6% for 84 days	80-week-old laying hens	No effect on final body weight, ADFI, FCR, and egg production Increase in egg weight (4% dosage) and liver weight.	Kara et al. [105]
	2.5, 4.5, 5.5, and 7.5% from 14 to 42 days	11-day-old broiler chicks	Decrease in ADFI and FCR (5.5 and 7.5% dosage) No effect on ADG	Kumanda et al. [99]
	10% from 14 to 42 days	11-day-old broiler chicks	No effect on ADFI but decrease in final body weight, ADG, and hot carcass weight (untreated grape pomace) No effect on growth performance (treated grape pomace with PEG or enzyme supplementation)	Kumanda and Mlambo [46]
	1.5% for 25 days	10-day-old female broilers	No effect on final body weight and FCR	Lichovnikova et al. [106]
	2.5% for 42 days	1-day-old broiler chicks	No effect on ADFI, final body weight, and FCR	Mavrommatis et al. [107]
	1 and 2% for 40 days	1-day-old broiler chicks	No effect on final body weight and ADG Decrease in body weight (1% dosage on d14) and FCR (non-statistical analysis)	Pascariu et al. [108]
	1, 2, and 3% for 35 days	74-week-old laying hens	Increase in ADFI (1–3% dosage) and egg production (1% dosage) Non-significant increase in FCR (3% dosage)	Reis et al. [109]
	3 and 6% for 28 days	50-week-old laying hens	Decrease in ADFI, egg weight (3 and 6% dosage), and FCR (6% dosage) No effect on egg production	Romero et al. [110]
	6% for 21 days	1-day-old male broiler chicks	No effect on final body weight and ADFI Decrease in FCR	Viveros et al. [43]
Grape pomace (polyphenolic extract)	15 mL/L (drinking water) for 40 days	1-day-old broiler chicks	No effect on final body weight and ADG Decrease in body weight on d28 Increase in FCR (non-statistical analysis)	Pascariu et al. [108]
Grape pomace (fermented and unfermented)	1.5% for 42 days	1-day-old female broiler chicks with 37.3 g	No effect on ADFI and FCR Increase in final body weight (fermented grape pomace)	Gungor et al. [44]
Grape seeds (fermented and unfermented)	0.5% for 42 days	1-day-old female broiler chicks with 37.3 g	No effect on ADFI and FCR Increase in final body weight and ADG	Gungor et al. [20]
Grape seeds	0.5, 1, and 1.5% for 84 days	44-week-old laying hens	No effect on ADFI and FCR Decrease in egg weight (1.5% dosage) Increase in egg production	Kaya et al. [111]
	0.5 and 1% for 40 days	1-day-old broiler chicks	Increase in final body weight (0.5% dosage) Decrease in body weight (1% dosage, d7–28) and FCR (non-statistical analysis) No effect on ADG	Pascariu et al. [108]
	2% for 5 weeks	14-day-old broilers with 312 g	No effect on final body weight, ADG, ADFI, and FCR	Turcu et al. [112]
Grape-seed extract	1, 2, and 4% for 42 days	1 day-old mixed-sex broiler chicks with 44.1 g	No effect on ADFI Increase (1 and 2% dosage) or decrease (4% dosage) in final body weight and ADG Decrease (2% dosage) or increase (4% dosage) in FCR	Abu Hafsa and Ibrahim [96]
	0.01 and 0.02% for 42 days	1 day-old female Pekin ducklings with 52.0 g	No effect on ADFI and FCR Increase in final body weight, ADG and carcass weight Increase in FCR	Ao and Kim [98]
	0.0025, 0.025, 0.25, and 0.5% for 21 days	1-day-old male broiler chicks	No effect on ADFI Increase in FCR and decrease in final body weight (0.5% dosage)	Chamorro et al. [113]
	0.0125, 0.025, 0.050, 0.100, and 0.200% for 42 days	1-day-old broiler chicks	No effect on growth performance	Farahat et al. [114]
	1% for 36 weeks and 2% for the last 2 weeks	4-week-old female broiler breeders	Decrease in final body weight and back fat thickness and increase in egg weight (2% dosage)	Grandhaye et al. [115]
	0.015, 0.03, and 0.045% for 42 days (heat stress at d29–42)	1-day-old broiler chicks	Increase in ADFI (0.03% dosage, d1–28) Increase in ADG (0.015 and 0.03% dosage on d1–28; 0.03% on d29–42 and overall period) Decrease in FCR (0.03 and 0.045% dosage on d29–42 and overall period)	Hajati et al. [116]
	0.1, 0.2, and 0.4% for 42 days	0-day-old broiler chicks	No effect on ADFI, final body weight, ADG, and FCR	Huerta et al. [117]
	0.0675, 0.1350, and 0.2025 for 84 days	44-week-old laying hens	No effect on ADFI and FCR Decrease in egg weight (0.0675% dosage) Increase in egg production	Kaya et al. [111]
	0.05 and 0.1% for 28 days	4-week-old laying hens	Decrease in ADFI (0.05 and 0.1%), egg weight, and FCR (0.05% dosage) No effect on egg production	Romero et al. [110]
	0.015% for 48 days	25-week-old laying hens	No effect on ADFI and FCR Decrease in egg weight	Sun et al. [118]
	0.72% for 21 days	1-day-old male broiler chicks	No effect on ADFI Decrease in ADG and, in comparison with antibiotic treatment, increase in FCR	Viveros et al. [43]
Grape seed (proanthocyanidin extract)	0.0005, 0.0010, 0.0020, 0.0040, and 0.0080% for 15 days (infection with *Eimeria tenella*)	1-day-old broiler chicks	Increase in ADG in comparison with infected control; decrease in ADG relative to noninfected control	Wang et al. [119]
	0.02 and 0.04% for 21 days	1-day-old broiler chicks	Decrease in ADFI and FCR Increase in ADG (0.02% dosage)	Cao et al. [120]
Grape skin (fermented and unfermented)	3 and 6% for 21 days	1-day-old male broiler chicks	No effect on ADFI Decrease in ADG (fermented and unfermented at 6%) Increase in FCR (fermented at 6% and unfermented at 3 and 6%)	Nardoia et al. [38]
Grape seed and skin (polyphenolic extract mixed with L-arginine, L-threonine and L-glutamine)	0.1% (5% grape extract and 95% amino acid mixture) for 35 days (challenge with coccidiosis vaccine on d14)	0-day-old male broiler chicks	The ADFI, final body weight, ADG, and FCR returned to control values	Chalvon-Demersay et al. [121]
Grape stems (pure phenolic extract)	0.1% for 42 days	1-day-old broiler chicks with 44.4 kg	No effect on ADFI, final body weight, and FCR	Mavrommatis et al. [107]

ADG, average daily gain; ADFI, average daily feed intake; FCR, feed conversion ratio.

## 5. Conclusions and Future Perspectives

Grape by-products have several industrial applications, including feed, biofuel, bioenergy, and fertilization/They are valuable feedstuffs due to their richness in nutritional and bioactive compounds, such as dietary fiber and polyphenols, which make them suitable for maintaining or improving animal performance. Indeed, conversely to the anti-nutritional properties found with high amounts of these compounds, low amounts were shown to modulate intestinal morphology and microbiota and stimulate antioxidant capacity, thereby maintaining intestinal health and preventing the occurrence of diseases in monogastric animals. Grape pomace can improve growth performance in pigs, with an increase in ADG, particularly when fed at higher levels (up to 9%). However, in poultry, the effect of grape by-products is more variable, and these sources should not be incorporated in broiler diets at more than 6–10% feed to prevent an impairment of animal growth performance. Forthcoming studies should concentrate their efforts on the optimization of dosages and digestibility of grape by-products for pig and poultry feeding.

## Figures and Tables

**Figure 1 animals-12-02239-f001:**
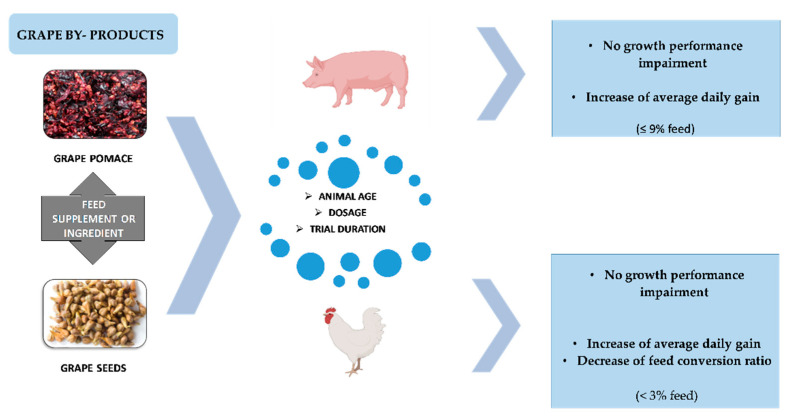
Grape by-products as feedstuff for pigs and poultry and consequences on animal production performance.

## Data Availability

The data presented in this study are available in this article.

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
