# Peer review of "Grape By-Products as Feedstuff for Pig and Poultry Production"

_animals, 2022, doi:10.3390/ani12172239_

Round 1

Reviewer 1 Report

Please find my comments to the attached document.

Author Response

  1. L11: Why only focusing in (limiting to) Mediterranean area (55/300 million hectoliters wine)? Are grape by-products different in other world-areas? South America (Chile, Argentina), US (California), South Africa, India, Australia are also main grape/wine producers.

Reply: The reviewer is right since we should have considered other world-areas and not limit our considerations to Mediterranean area. “The Mediterranean countries” was removed from line 11. This aspect is now considered in the Introduction section (L59-84). However, the statistics about wine production in 2021 indicate that Mediterranean countries (Italy, France, Spain and Portugal) produced about 130.4 million hectoliters wine (and not 55 million hl) in total of 260 million hl, during this period (OIV 2022, State of the World and Vine and Wine Sector). Therefore, the production in these European countries is about half of the worldwide production.

  1. L13: Why is there no reference for their use in ruminants (dairy cows, goat, sheep), rabbits or laying hens that can tolerate higher Cfibre and ANF better? Please reference grape by-product use in these animal categories.

Reply: We thank the reviewer suggestion, but the purpose of our study was to analyze the use of grape by-products in monogastric animals, particularly broilers and pigs, and not in ruminants, since the meat of these animals in the most consumed worldwide. Although rabbits are also monogastric animals, they correspond to a smaller proportion of meat. For a better clarification of this aspect, we replaced “monogastric animals” by “pigs and poultry”, in lines 13 to 14 and 101. The studies found for laying hens are now added in Table 2 to avoid limiting the review to broiler chickens and due to the recognized importance of eggs for consumption.

  1. Potential use as fertilizers, and other uses? Please include them as well in the Introduction/Discussion.

Reply: We acknowledged the reviewer suggestion and mentioned the potential use of grape by-products as fertilizers and others in the Introduction (lines 87 to 92) and Conclusion (lines 301 to 302) sections.

  1. L29: Poultry or broilers?

Reply: Poultry. The review was extended to laying hens and not only broilers. Besides that, one study in cockerels and another in ducks were previously presented in the Results section (see Table 2).

  1. L34-35: poultry: same levels need to achieve same results. Which results? increase ADG or maintain performance?

Reply: Lower levels of by-products are needed for poultry to achieve an increase of ADG compared to pigs. The sentence was modified in lines 33 to 34.

  1. L49: and milk and eggs...

Reply: The eggs are now considered in line 49. Although milk is also an animal product with high relevance for consumption, we decided to keep this review within the monogastric animals and not ruminants.

  1. L68: same comment as previously indicated: why so much focus on the Mediterranean if it represents 1/3 of grape production and grape by-products should be the same in other world areas (?)

Reply: We thank the reviewer for the pertinent comment. This aspect is now considered between lines 60 and 85.

  1. why 20 years? Has there been a critical technological change on wine production that originated different composition of existing by-products or even new by-products?

Reply: The studies done in the last decades were selected because the present report consists of an updated systematic review about the use of grape by-products for the monogastric animals (line 107). To the best of our knowledge, there was no critical technological change related with by-products´ production, but an increased awareness of the importance of a sustainable agriculture and animal feeding to face the climate change occurred in the beginning of the 21st century (lines 87 to 88).

  1. L105: I would suggest including other terms such as "grape pomace" and other grape by-product names. That may be the reason for the limited number of papers found. Also, more specific animal categories should be included such as "broiler chickens, "laying hens", "weaned piglets", etc.

Reply: Thank you for your suggestion. We repeated our literature search with the terms "grape", "by-products", "hens" and "weaned piglets" and found more studies, which are included in Table 2. The keywords used for research were changed between lines 118 and 119, accordingly.

  1. L116-119: It is important to mention that Grape pomace can include seeds or be seedless. Relevant to make a distinction between studies testing pomace including, or not, seeds?

Reply: The distinction between whole grape pomace, seedless and seeds is now presented between lines 133 and 146. The studies testing grape pomace on animals used whole grape pomace (with seeds and not seedless).

  1. L138: I can´t see Table 1?

Reply: Table 1 is now added to the manuscript.

  1. L140: Potential use of phytase on P digestibility?

Reply: The sentences about the potential use of phytase are now added between lines 173 and 175 and lines 198 to 204.

  1. L229, Table 2: Not clear to what grape by-product each reference belongs to.

Reply: Thank you. Now the references were better clarified in Table 2.

  1. L286, Table 3: Same comment as in Table 2.

Reply: Done.

Reviewer 2 Report

Dear Authors,

1. It mentioned many times in the texts that Grapes products contains Polyphenols, it might be good to provide a few lines and explain the benefits of the compounds and possibly relate them to poultry or pigs production before section 4.1 and 4.2. 

2. Line 193-197. I would recommend you to provide 2-3 citations for poultry and pig studies here. 

3. The conclusion section is wordy and it can be shorter with the same outcomes. Please avoid repetitive statements and focus on key words. 

Regards, 

Author Response

Dear Authors,

  1. It mentioned many times in the texts that Grapes products contains Polyphenols, it might be good to provide a few lines and explain the benefits of the compounds and possibly relate them to poultry or pigs production before section 4.1 and 4.2. 

Reply: Thank you for your comment. The benefits of polyphenols are now better explained between lines 243 and 269 (section 4).

  1. Line 193-197. I would recommend you to provide 2-3 citations for poultry and pig studies here. 

Reply: The citations are now provided between lines 278 and 280.

  1. The conclusion section is wordy and it can be shorter with the same outcomes. Please avoid repetitive statements and focus on key words. 

Reply: The conclusion section was reduced, and repetitive statements avoided.

Reviewer 3 Report

This review is really interesting when the potential of using grape by-product is considerable. However, the authors need to improve the content and make clear in some points based on my comments as follow:

- In section 3, it would be better if the authors presented the nutritional composition of Grape By-Products in a table which needs to contain basic proximate nutritional analysis (moisture, crude protein, energy...)

- It also must present any harmful or toxic substance incorporated in this by-product and their effect on animals health.

- Benefit of using grape by-products in animal diets: improved performance, meat quality, health, or reduce feed cost, or environmental benefit?

- The level of grape by product in the diet need to be clear they were fresh or dried basis?

Author Response

This review is really interesting when the potential of using grape by-product is considerable. However, the authors need to improve the content and make clear in some points based on my comments as follow:

  1. In section 3, it would be better if the authors presented the nutritional composition of Grape By-Products in a table which needs to contain basic proximate nutritional analysis (moisture, crude protein, energy...)

Reply:  The nutritional composition of grape by-products in now presented in Table 1.

  1. It also must present any harmful or toxic substance incorporated in this by-product and their effect on animals’ health.

Reply: Harmful and toxic substances incorporated in grape by-products and their potential effects on animal health are now specified in Table 1 and between lines 158 and 183

  1. Benefit of using grape by-products in animal diets: improved performance, meat quality, health, or reduce feed cost, or environmental benefit?

Reply: Than you for your comment. Those aspects are now added between lines 196 and 202 and lines 206 and 212. The effect of grape by-products on meat quality will soon be published by a companion paper.

  1. The level of grape by product in the diet need to be clear they were fresh or dried basis?

Reply: The level of grape by product was added in the diet as dried basis. That aspect is now specified in Tables 2 and 3 and in lines 279, 307 and 343.

Round 2

Reviewer 1 Report

Dear authors,

The manuscript looks better than the previous version. Here a couple more comments:

1- Table 1: Could you add energy values (either GE, ME, NE) and digestible AA values for the different grape by-products? This could be done in a separate table.

2- As shown in the first review, literature search could be improved. There is still room for improvement by adding many other keywords: "laying hens", "laying hen pullets", "turkeys", "broiler", "broiler chicken", "weaned pigs", "weaning", "grape pomace", "seedless pomace", "grape seed", etc. This point is critical to conduct a correct  literature review, and have the manuscript accepted.

Author Response

Dear authors,

The manuscript looks better than the previous version. Here a couple more comments:

  1. Table 1: Could you add energy values (either GE, ME, NE) and digestible AA values for the different grape by-products? This could be done in a separate table.

Reply: Thank you for your comment. The energy values (ME) and AA values for the grape by-products were added to Tables 1 and 2.

  1. As shown in the first review, literature search could be improved. There is still room for improvement by adding many other keywords: "laying hens", "laying hen pullets", "turkeys", "broiler", "broiler chicken", "weaned pigs", "weaning", "grape pomace", "seedless pomace", "grape seed", etc. This point is critical to conduct a correct literature review, and have the manuscript accepted.

Reply: We acknowledge the reviewer suggestion and improved the literature search (please see updated Tables 3 and 4). New keywords were added between lines 103 and 112, page 3.